# mTORC2 Is the Major Second Layer Kinase Negatively Regulating FOXO3 Activity

**DOI:** 10.3390/molecules27175414

**Published:** 2022-08-24

**Authors:** Lucia Jimenez, Carlos Amenabar, Victor Mayoral-Varo, Thomas A. Mackenzie, Maria C. Ramos, Andreia Silva, Giampaolo Calissi, Inês Grenho, Carmen Blanco-Aparicio, Joaquin Pastor, Diego Megías, Bibiana I. Ferreira, Wolfgang Link

**Affiliations:** 1Institute of Biomedical Research Alberto Sols (CSIC-UAM), Arturo Duperier 4, 28029 Madrid, Spain; 2Fundación MEDINA, Health Sciences Technology Park, Avda. del Conocimiento 34, 18016 Granada, Spain; 3ABC-RI, Algarve Biomedical Center Research Institute, Algarve Biomedical Center, 8005-139 Faro, Portugal; 4Faculty of Medicine and Biomedical Sciences, University of Algarve, 8005-139 Faro, Portugal; 5Spanish National Cancer Research Centre (CNIO), Melchor Fernández Almagro 3, 28029 Madrid, Spain

**Keywords:** FOXO, kinases, mTOR, high content screening, chemical biology, cancer, aging

## Abstract

Forkhead box O (FOXO) proteins are transcription factors involved in cancer and aging and their pharmacological manipulation could be beneficial for the treatment of cancer and healthy aging. FOXO proteins are mainly regulated by post-translational modifications including phosphorylation, acetylation and ubiquitination. As these modifications are reversible, activation and inactivation of FOXO factors is attainable through pharmacological treatment. One major regulatory input of FOXO signaling is mediated by protein kinases. Here, we use specific inhibitors against different kinases including PI3K, mTOR, MEK and ALK, and other receptor tyrosine kinases (RTKs) to determine their effect on FOXO3 activity. While we show that inhibition of PI3K efficiently drives FOXO3 into the cell nucleus, the dual PI3K/mTOR inhibitors dactolisib and PI-103 induce nuclear FOXO translocation more potently than the PI3Kδ inhibitor idelalisib. Furthermore, specific inhibition of mTOR kinase activity affecting both mTORC1 and mTORC2 potently induced nuclear translocation of FOXO3, while rapamycin, which specifically inhibits the mTORC1, failed to affect FOXO3. Interestingly, inhibition of the MAPK pathway had no effect on the localization of FOXO3 and upstream RTK inhibition only weakly induced nuclear FOXO3. We also measured the effect of the test compounds on the phosphorylation status of AKT, FOXO3 and ERK, on FOXO-dependent transcriptional activity and on the subcellular localization of other FOXO isoforms. We conclude that mTORC2 is the most important second layer kinase negatively regulating FOXO activity.

## 1. Introduction

FOXO proteins are transcription factors responsible for the maintenance of cellular homeostasis [1]. They belong to the family of Forkhead proteins, characterized by a ~100-residue forkhead (FKH) DNA-binding domain [2] that, in mammals, consists of FOXO1, FOXO3, FOXO4 and FOXO6 [3]. FOXO transcription factors bind as monomers to consensus binding sites within the promoter of their target genes. The growing list of established FOXO target genes includes genes involved in cell proliferation, metabolism, apoptosis, autophagy and stress resistance [4]. FOXO proteins are tumor suppressors frequently inactivated in human cancer [5]. On the one hand, many anti-cancer drugs act through FOXOs and their inactivation is a powerful mechanism of therapy resistance [6,7]. On the other hand, several transcriptional targets of FOXO factors have been shown to promote drug resistance or to be implicated in feedback loops [7]. Furthermore, genetic variants of FOXO3 are associated with exceptional longevity in worms, flies and mammals. In humans, FOXO3 hosts about 40 common non-coding single nucleotide polymorphisms (SNPs) that have been consistently associated with longevity [8]. FOXOs are mainly regulated by reversible post-translational modifications (PTMs) which generate a molecular code to sense external stimuli and determine the transcriptional programs mediated by these transcription factors [9]. The main regulatory input to FOXOs activity comes from growth factor-dependent and stress signaling [1]. The PTMs determine activity of FOXO proteins by regulating their subcellular localization, protein stability and transcriptional activity. Under stress conditions or in the absence of growth or survival factors, FOXO proteins translocate to the cell nucleus, where their transcriptional functions can be executed. Conversely, in the presence of growth factors or in cancer cells where the PI3K/AKT pathway is constitutively activated, AKT phosphorylates FOXO transcription factors in the nucleus, creating a docking site for the 14-3-3 protein dimer. The binding of the 14-3-3 chaperone to nuclear FOXO reduces its affinity to DNA and facilitates its nuclear export eventually leading to cytoplasmic sequestration and inactivation of FOXO. Several kinases have shown direct phosphorylation of FOXO proteins including AKT, SGK, IKK, ERK, MST1 and JNK [10,11,12,13,14], which we refer to as the first layer of regulation of FOXO functions by kinases. Furthermore, kinases acting upstream of these kinases represent a second layer of control. Here we investigate the inhibition of several second layer kinases including PI3K, mTOR and MEK, as well as ALK and other receptor tyrosine kinases (RTKs) on the subcellular localization and transcriptional activity of FOXO3. RTKs are cell surface receptors activated by growth factors to produce a downstream response that includes the activation of the PI3K/AKT signaling cascade [15]. A wide variety of RTK inhibitors have been approved for clinical use by regulatory authorities. PI3Ks function as heterodimers consisting of one of four catalytic p110 subunits. ATP competitive inhibitors have been developed that are capable of inhibiting the kinase activity of p110α, β, δ and γ or in an isoform specific manner [16]. The serine/threonine protein kinase mTOR is part of the PI3K/AKT signaling network and represents the catalytic subunit of two distinct protein complexes, known as mTOR Complex 1 (mTORC1) and 2 (mTORC2) [17]. Here, we show that pharmacological manipulation of the different second layer regulatory kinases has differential effects on the intracellular localization of FOXO3 suggesting a complex regulation of FOXO proteins.

## 2. Results

### 2.1. FOXO3 Accumulates in the Nucleus upon the Inhibition of Regulatory Kinases

In order to explore if inhibition of second layer regulatory kinases efficiently induces the nuclear shuttling of FOXO3 transcription factor, we treated a previously established reporter cell line, U2foxRELOC, with small molecule kinase inhibitors. U2foxRELOC stably expresses a fluorescently labeled FOXO3 fusion protein and enables an image-based approach to monitor the subcellular localization of FOXO3, making it compatible with high-throughput evaluation of small molecule compounds [18]. In order to rule out secondary effects on FOXO proteins, e.g., processes that involve transcription and de novo synthesis of proteins, we limited the incubation period to one hour. We treated these reporter cells with small molecule inhibitors of secondary regulatory kinases including PI3K, mTOR, EGFR, HER2, VEGF, ALK, ROS, MET and MEK. Table 1 lists the compounds used in our experiments.

As all compounds have been dissolved in DMSO, we used DMSO as a negative control. The maximum concentration of DMSO used in the experiment was 0.5% to avoid vehicle-mediated toxicity that could interfere with the assay. Initial experiments were carried out at a single concentration in quadruplicate. After drug incubation, cells were fixed, and their nuclei stained with Hoechst 33342 dye. Images were acquired by fluorescent microscopy and image analysis was performed with specialized software. Images were analyzed at single cell level quantifying the green fluorescence within the area stained with Hoechst 33342, which defined the cell nucleus, and the extended cytoplasm. As shown in Figure 1, in the presence of 0.5% DMSO the fluorescent signal was mostly distributed in the cytoplasm (Figure 1A). The treatment of the cells with the nuclear export inhibitor leptomycin B (LMB) at a concentration of 20 nM shifted the reporter protein almost entirely into the cell nucleus (Figure 1B). As previously demonstrated, the PI3K inhibitor LY294002 also induced nuclear FOXO translocation [19]. We used the nuclear translocation of FOXO3 upon the treatment with LY294002 at 25 μM as a reference to quantify the effect of other inhibitors of the second layer kinases (Figure 1C). When compared to the pan-PI3K inhibitor LY294002, idelalisib, a selective, FDA-approved p110δ inhibitor with a reported EC50 of 2.5 nM in cell-free assays, exhibited significantly less activity on FOXO3 translocation (Figure 1D). This observation suggests that p110δ contributes less than other p110 isoforms to AKT-mediated FOXO3 inactivation in osteosarcoma cells. Furthermore, when cells were treated with PIK-75, a selective p110α inhibitor, the nuclear translocation of FOXO is observed even at low concentrations (Figure 1E). While the FDA-approved ALK/MET/ROS1 inhibitor crizotinib shifted the fluorescent reporter signal to the cell nucleus at an EC50 value of 5 μM (Figure 1M), it also induced acute toxicity at higher concentrations. This data suggests that the receptor tyrosine kinases (RTKs) ALK, MET or ROS1 represent upstream regulatory components for FOXO activity. Conversely, the inhibition of the RTKs VEGFR1, VEGFR2 and VEGFR3 by lenvatinib, EGFR/HER2 by the reversible dual inhibitor lapatinib and EGFR with the reversible or irreversible inhibition of EGFR by erlotinib or afatinib, respectively, failed to affect the subcellular localization of FOXO3 (Figure 1G–J). Similarly, inhibition of MEK with the FDA-approved inhibitor trametinib had no effect on the subcellular localization of FOXO3 (Figure 1K). Furthermore, all compounds were tested in dose-response experiments using 12 serial dilutions (Appendix A). These results suggest that HER2, EGFR and MEK are not significantly involved in the regulation of FOXO3 localization in osteosarcoma cells. As MEK represents the major upstream regulator of ERK and ERK has been reported to regulate FOXO3 translocation by direct phosphorylation [12], we explored the effect of inhibiting ERK using the ATP competitive inhibitor of ERK1/2, SCH772984. Intriguingly, SCH772984 also failed to affect FOXO subcellular localization (Figure 1L).

### 2.2. Lack of Effect of MEK and ERK Inhibitors Is Not Cell Type Specific

The absence or low expression level of a molecular target may significantly affect the response to drug treatment. In order to investigate if the lack of response to RTK, MEK and ERK inhibition can be explained by the level of expression of their molecular targets, we assessed the Human Protein Atlas database in order to analyze the expression of VEGFR1, VEGFR2, VEGFR3, EGFR, HERr2, MEK and ERK1/2 in bone tissue and U2OS osteosarcoma cells. While VEGFR1, VEGFR2, EGFR and HER22 indeed exhibit zero or low expression in U2OS cells at transcript level, transcripts of VEGFR3, MEK and ERK1/2 were readily detectable in U2OS cells. No protein data is available for U2OS cells. In order to explore the possibility that the inactive inhibitors affect FOXO3 in other cell lines we used immunocytochemistry and reporter cell lines. We treated human derived neuroblastoma cells (SH-SY5Y) and a human neonatal dermal fibroblast cell line (CC-2509) with 1μM trametinib, SCH772984 and lenvatinib for one hour. As Figure 2 shows, none of the inhibitors cause FOXO3 translocation into the cell nucleus (Figure 2E–J) compared to LY294002 (Figure 2C,D), showing that the most relevant pathway in FOXO3 regulation is PI3K/AKT. It is important to note that trametinib and SCH772984 are known to affect the activity of the MAPK pathway in U2OS cells [20].

In order to investigate whether inhibition of the MAPK pathway could affect the subcellular localization of FOXO3 in a context in which this pathway is activated, we monitored FOXO3 localization after EGF treatment and subsequent trametinib exposure in U2foxRELOC cells (Appendix A). In addition, we used UACC62 melanoma cells that carry a BRAF mutation and therefore exhibit constitutively active MAPK signaling. Trametinib treatment failed to induce nuclear FOXO3 localization in both cellular models (Appendix A), indicating that MEK inhibition did not affect FOXO3 trafficking in cells with either acute or constitutive activation of the MAPK pathway.

### 2.3. Inhibition of mTOR Kinase Induces Nuclear FOXO3 Accumulation

Next, we investigated the contribution of mTOR activity and, in particular, the role of the two mTOR containing complexes mTORC1 and mTORC2 in the regulation of FOXO3. We found that the dual PI3K/mTOR inhibitors dactolisib and PI-103 were more efficient FOXO3 translocators than specific PI3K inhibitors (Figure 3B,C). We also observed that the ATP-competitive mTOR inhibitor torin-1 potently shifts the fluorescent signal into the cell nucleus without any traces left in the cytoplasm (Figure 3D). To assess whether other specific mTOR inhibitors affect FOXO3 localization with similar potency, we treated the reporter cells with KU-0063794, sapanisertib and AZD8055 (Figure 3E–G). Intriguingly, all inhibitors of mTOR kinase activity exhibited very low EC50 values (Figure 3J and Appendix A). The most potent inhibitor was AZD8055, affecting nuclear FOXO3 translocation at an EC50 of 9 nM (Figure 3J).

In order to exclude the possibility that these inhibitors affect the nuclear export of FOXO3 mediated by the export receptor CRM1 (Chromosomal Maintenance 1, also known as Exportin 1), which is the major mammalian export protein that facilitates the transport of large macromolecules, we used a previously established reporter system capable of identifying small molecule inhibitors of the nuclear export [18]. In a multiplexed assay, we co-cultured U2redNES and U2foxRELOC cells and treated cells with 100nM of torin-1 for 1 h using leptomycin B as a positive control (Figure 3H). As shown in Figure 3D, torin-1 does not affect the nuclear export through CRM1 indicating that mTOR inhibition acts through inhibiting regulatory components operating upstream of FOXO. We then determined if there is a dose response relationship between mTOR inhibition and FOXO3 nuclear translocation (Appendix A).

### 2.4. mTORC1 Inhibition Does Not Affect FOXO3 Localization

As torin-1, KU-0063794, sapanisertib and AZD8055 are ATP-competitive inhibitors of the catalytic subunit of mTOR kinase, they affect the activity of the two mTOR complexes mTORC1 and mTORC2. Therefore, we investigated the contribution of each mTOR complex to the nuclear translocation of FOXO3 using rapamycin. Rapamycin is an extremely selective compound that binds to a domain separate from the catalytic site and specifically inhibits mTORC1 with an EC50 value in the high pM/low nM range. The treatment of U2foxRELOC cells with 0.1 nM of rapamycin had no effect on the intracellular distribution of the green fluorescent signal (Figure 3I). In order to rule out that there was a dose effect, we used higher concentrations of rapamycin. We observed that concentrations of rapamycin as high as 100 nM did not affect the subcellular localization of FOXO3, suggesting that the mTORC1 does not contribute to the regulatory input of mTOR in FOXO activity (Figure 3J). In order to explore the possibility that this specific inhibition was due to a cell-specific context, we used immunocytochemistry and reporter cell lines CC-2509 and SH-SY5Y. Cell lines were treated with 1 μM KU-0063794, sapanisertib, AZD8055 and rapamycin for 1 h prior to cell fixation and immunostaining. As shown in Figure 4, only the ATP-competitive inhibitors of the two mTOR complexes shifted FOXO3 into the cell nucleus in both cell lines while rapamycin did not affect FOXO subcellular localization, reproducing the result obtained in U2OS cells. Furthermore, the data suggests that the potent effect of the ATP-competitive mTOR inhibitors on the subcellular localization of FOXO3 is exclusively mediated by mTORC2.

### 2.5. Second Layer Kinase Inhibitors Differentially Affect FOXO and AKT Phosphorylation

FOXO3 is inactivated by phosphorylation of three conserved sites via the serine/threonine kinase AKT, which in turn is regulated by phosphorylation of its threonine 308 and serine 473 residues. The serine 473 in AKT is known to be phosphorylated by mTORC2 [21]. To investigate if the effect or the lack of effect of second layer kinase inhibitors on FOXO nuclear translocation correlates with FOXO and AKT phosphorylation, we treated U2OS cells with 1 μM PIK-75, idelalisib, dactolisib, PI103, torin 1, KU-0063794, sapanisertib, AZD8055 and rapamycin. As shown in Figure 5A, western blot analysis using specific antibodies against FOXO3-S253, AKT-S473 and total AKT confirmed that treatment of U2OS cells with PI3K and mTOR inhibitors, except for rapamycin and idelalisib, efficiently reduces phosphorylation of AKT at S473 and phosphorylation of FOXO3 at S253. Furthermore, the FOXO and AKT phosphorylation status in U2OS treated with the RTKs and MEK/ERK inhibitors were unchanged (Appendix A). These results indicate that the effect of the analyzed compounds on FOXO translocation depend on their efficacy to decrease the levels of AKT and FOXO.

### 2.6. Inhibition of PI3K and mTOR Drive FOXO-Dependent Transcriptional Activity

As we demonstrated that PI3K and mTOR inhibition leads to FOXO accumulation in the nucleus, we carried out gene reporter assays in order to determine if the treatment of cells with PI3K and mTOR inhibitors increased transcriptional activity mediated by FOXO proteins. Following the manufacturer’s instructions, the Dual-Luciferase Reporter Assay was carried out with PI3K and mTOR inhibitors at 1 μM, DMSO as the reference control and 25 μM LY294002 as the positive control, and results are represented in Figure 5B. The PI3Kδ inhibitor idelalisib and ALK/MET/ROS1 inhibitor crizotinib only slightly increased the FOXO dependent generation of luciferase expression. In line with the data obtained from the dose-response experiments on FOXO translocation, the most potent inducers of FOXO-dependent gene transcription were the PI3K and mTOR inhibitors, and in particular the specific inhibitors of the mTOR kinase activity. As for the nuclear translocation of FOXO, the ATP-competitive mTOR inhibitor AZD8055 was the most potent at inducing FOXO-dependent transcription. These results suggest that there is a direct relationship between the amount of nuclear FOXO factors and their transcriptional activity as the inhibitors that most efficiently translocate FOXO3 also exhibited the strongest induction of luciferase reporter gene expression.

### 2.7. PI3K and mTOR Inhibition Affects the Localization of FOXO Isoforms FOXO1 and FOXO4

In mammals, four FOXO isoforms have been identified: FOXO1, FOXO3, FOXO4 and FOXO6. FOXO6 has been shown to be less responsive to regulation by its subcellular localization [22]. As our reporter cell line only monitors the translocation of FOXO3, we investigated if PI3K and mTORC inhibition also induces the nuclear accumulation of FOXO1 and FOXO4. To this end, we carried out immunofluorescence assays with U2OS cells. We decided to treat the cells with the compounds at a concentration of 1 μM to explore if FOXO1 and FOXO4 isoforms were as responsive as FOXO3. Then, cells were fixed, permeabilized and incubated with specific antibodies against FOXO1, FOXO3 and FOXO4 isoforms. Cells were observed by fluorescence microscopy after the incubation with a green fluorescent-labeled secondary antibody and DAPI to visualize the nuclei. As shown in Figure 6, compared to DMSO vehicle control, treatment with the positive control LY294002 as well as with 1 μM of idelalisib, crizotinib or AZD8055 led to the accumulation of FOXO1, FOXO3 and FOXO4 isoforms in the cell nucleus. It is important to note that with this approach we monitor the response of endogenous FOXO isoforms, including FOXO3, confirming the data obtained from experiments with the reporter cell line expressing ectopic GFP-FOXO3. Interestingly, the compounds induce the nuclear translocation of FOXO1 and FOXO4 as potently as for FOXO3. In agreement with previous data, treatments with idelalisib and crizotinib were the least responsive ones (Figure 6G–L), confirming that p110δ and ALK/MET/ROS1 are not a prevalent regulatory input of endogenous FOXO3 nor of FOXO1 and FOXO4.

## 3. Discussion

Given the role of FOXO factors in pathological processes and human longevity, FOXO proteins have emerged as promising targets to treat cancer, overcome therapy resistance and slow aging [23]. As other non-liganded transcription factors, FOXO proteins are considered to be difficult to target by small molecule compounds [24]. While progress has been made in targeting transcription factors [25], and in particular FOXOs [26], by agents that directly bind to these factors, the upstream regulatory network provides a broad range of intervention points to pharmacologically modulate FOXO activity. Therefore, a better understanding of the regulation of FOXO proteins will allow the identification of the most efficient upstream targets. In order to understand the regulation of the subcellular localization and transcriptional activity of FOXO factors, we have previously investigated the effect of inhibiting kinases that directly phosphorylate FOXO, denominated first layer kinases [18,19]. Here, we have assessed the effect of inhibiting different kinases that act upstream of these first layer kinases, denominated as second layer kinases on the subcellular localization and transcriptional activity of FOXO factors and in particular FOXO3. With this chemical biology approach, we determined the contribution of these second layer kinases to the regulation of FOXO3. We show that while the inhibition of EGFR has little effect on the subcellular localization of FOXO3, the ALK/MET/ROS1 inhibitor crizotinib induces the nuclear translocation of FOXO3. HER2 and VEGF inhibition via lapatinib and lenvatinib, respectively, did not induce the nuclear accumulation of FOXO3. As the compounds we used to inhibit EGFR are drugs approved for their clinical use, the lack of regulatory activity on FOXO3 is likely due to reasons other than cell penetration or intracellular stability. Conversely, the level of expression of the targets in the context of osteosarcoma-derived cells might limit their effect on FOXO3. Alternatively, other RTKs might have a more significant role in regulating FOXO subcellular localization even in the presence of EGFRs. Accordingly, crizotinib, an FDA-approved drug used for the treatment of patients with ALK-positive and ROS1-positive non-small cell lung carcinoma (NSCLC), triggered the nuclear translocation of FOXO3 suggesting an important role of RTKs such as ALK, MET or ROS1 in the regulation of FOXO3. Which of these RTKs is the major mediator of this effect remains to be established. Furthermore, we investigated the effect of MEK inhibition on FOXO3 regulation. Importantly, the MAPK/ERK pathway has shown to play an important role in the regulation of FOXO3 [12]. The serine/threonine kinase ERK directly interacts with FOXO3 and phosphorylates the protein at three serine residues which is followed by MDM2-mediated ubiquitin-proteasomal degradation of FOXO3 and subsequent cell proliferation and tumorigenesis. As ERK-mediated phosphorylation leads to nuclear exclusion of FOXO3 and ERK itself is regulated by the upstream kinase MEK, we reasoned that MEK inhibition also should result in FOXO3 nuclear accumulation. However, treatment of the reporter cells with the FDA-approved MEK inhibitor trametinib failed to have an effect on the subcellular localization of FOXO3, as well as ERK inhibition with SCH772984. While we cannot rule out that the cellular context in the osteosarcoma-derived cell line limits the impact of the MAPK/ERK pathway on FOXO3, the MAPK inhibitors used in the experiments have been seen to affect MAPK signaling in U2OS cells. The observation that the FDA-approved specific PI3Kδ inhibitor idelalisib triggered FOXO translocation activity suggests that the p110δ catalytic subunit of the PI3K contributes to the regulation of FOXO3 activation. However, p110δ is expressed primarily in hematopoietic lineage, and acts as an important regulator of normal and malignant B-cells [27]. The most potent small molecule compound tested in these series of experiments was the ATP-competitive mTORC1/2 inhibitor AZD8055 with an EC50 value in the low nanomolar range [28]. Another potent FOXO translocating compound was the imidazoquinoline dactolisib (BEZ235), which is known to inhibit p110α/γ/δ/β and mTOR in an ATP-competitive manner with an EC50 value in the low nanomolar range. The efficacy of dactolisib-mediated FOXO3 nuclear translocation could be due to the dual inhibitory activity against two second layer kinases [29]. While inactivation of PI3K leads to decreased levels of PIP3 and in turn less AKT localized at the plasma membrane, mTOR inhibition results in lower levels of AKT phosphorylation at Ser473. Both events may act synergistically to attenuate the activation of AKT and a decrease in AKT mediated FOXO phosphorylation. mTORC2 has been seen to contribute to the regulation of FOXO3 localization [30]. In accordance with these results, the ATP-competitive inhibitors of mTOR, torin1, KU-0063794, sapanisertib and AZD8055 efficiently induced the nuclear translocation of FOXO3 confirming that reduced phosphorylation at the mTORC2 site at Ser473 in AKT inhibits its enzymatic activity and AKT mediated FOXO phosphorylation. As a result, unphosphorylated FOXO can accumulate in the cell nucleus. The contribution of the other mTOR complex, mTORC1 to the regulatory input of FOXO activity is unclear. Our observation that the specific mTORC1 inhibitor rapamycin failed to induce FOXO nuclear translocation suggests that the potent effect of dactolisib is probably due to a synergistic effect of pan-PI3K and mTORC2 inhibition, without a significant effect on the mTORC1 complex. It is important to note that several of the small molecule compounds analyzed in the current study have been approved for their clinical use to treat patients with cancer. These drugs include afatinib, crizotinib, erlotinib, idelalisib, lapatinib, lnvatinib, rapamycin and trametinib. Our data show that crizotinib and idelalisib can induce the nuclear translocation of the three FOXO isoforms, FOXO1, 3 and 4, suggesting that FOXO activation might contribute to the therapeutic effect of these drugs. For the other approved inhibitors, it remains to be established whether they fail to induce FOXO activation in specific tissues or tumors in patients. In line with the data obtained from translocation experiments, the most potent compounds to reduce AKT levels and induce FOXO phosphorylation and FOXO-mediated gene transcription were dual PI3K/mTOR inhibitors and in particular, specific mTOR kinase inhibitors. Taken together, our results suggest that the regulatory input operating through the mTORC2 complex represents the major second layer control of FOXO activity.

## 4. Materials and Methods

### 4.1. Cell Culture

U2-OS and SH-SY5Y cells as well as CC-2509 fibroblasts were obtained from the American Type Culture Collection (ATCC) and were maintained in DMEM supplemented with 10% FBS (Sigma-Aldrich, St. Louis, MO, USA) and antibiotics (Gibco, Waltham, MA, USA). Cell cultures were maintained in a humified incubator at 37 °C with 5% CO_2_ and passaged when confluent using trypsin/EDTA. Stable cell lines U2redNES and U2foxRELOC cells have been generated as described previously [19,31,32]. Briefly, U2foxRELOC is a cell line derived from U2OS cells stably transfected with a reporter plasmid to express a FOXO3 fusion protein with green fluorescence protein (GFP) [19]. This allows to distinguish the subcellular localization of FOXO3 with fluorescence microscopy and to identify compounds that make FOXO accumulate in the nucleus. U2redNES is a cell line derived from U2OS cells stably transfected with a reporter plasmid to express red fluorescence protein dsRed with the NES from MAPKK (Jimenez et al., manuscript under revision). This sequence was chosen because it is necessary for the nuclear export by CRM1 as well as FOXO’s NES, so a compound that inhibits CRM1 like leptomycin B will retain both signals GFP and dsRed in the nucleus.

### 4.2. FOXO Translocation Assay in U2foxRELOC and U2redNES

The effect of the compounds on the subcellular localization of FOXO was shown by U2foxRELOC reporter cells, whereas U2redNES cells were used as a nuclear export integrity maintenance control. For this, 7000 U2foxRELOC and 13,000 U2redNES cells were seeded per well in 200 μL medium in a black well clear flat bottom TC-treated 96-well plate (Greiner, Frickenhausen, Germany) and incubated for 24 h. Compound treatments were carried out per quadruplicate in sterile conditions at 500 nM final concentration. In order to rule out secondary effects on FOXO proteins like transcription and de novo protein synthesis, the incubation period was limited to 1 h at 37 °C. As inhibitors are dissolved in DMSO, cells treated with 1% DMSO were used as negative control. Pan-PI3K inhibitor LY294002 (25 μM) was used as positive control for FOXO accumulation in the nucleus and CRM1 inhibitor Leptomycin B (20 nM) was used as positive control for the inhibition of nuclear export [33]. After the incubation period with compounds, cells were fixed using 4% formaldehyde (Sigma-Aldrich) dissolved in PBS 1X with 5 μg/mL Hoechst 33342 (Sigma-Aldrich) for nuclei staining during 20 min in the dark at RT. Subsequently, each well was washed with 100 μL PBS 1X. A total of 100 μL sodium azide (Sigma-Aldrich) 0.05% dissolved in PBS 1X was added and plates were sealed with Parafilm for storage at 4 °C [18,19,34]. Cells were analyzed using an optical microscopy, Cell Observer Z1 (Zeiss, Oberkochen, Germany), photographs were taken using a camera installed on the microscope, Prime BSI Express (Photometrics), and directed with the ZEN 3.3 software (Zeiss).

### 4.3. EGF Induction

U2foxRELOC were seeded on coverslips in 24-well plates at a concentration of 50,000 cells/wells. After 24 h cells were washed with DMEM and maintained 4 h in non-supplemented DMEM. Then cells were incubated for 1 h in a complete medium with 50 ng/mL EGF before exposure to 0.5% DMSO, 20 nM leptomycin B (LMB), 500nM dactolisib or 500nM trametinib. Finally, cells were fixed with 4% formaldehyde, washed with PBS, incubated with 5 µg/mL DAPI for 15 min and coverslips were mounted on a slide with ProLong. The photos were taken with a Nikon 90i at 20× (Nikon, Tokyo, Japan).

### 4.4. Dose-Response Assay in U2foxRELOC

U2foxRELOC cells, 7000 per well, were seeded in 60 μL medium in a black well with optically clear film bottom TC-treated MW384 plate (PerkinElmer, Waltham, MA, USA) and incubated for 24 h. Ten 1:2 serial dilutions were made with medium in a final volume of 4 μL, then 96 μL medium was added to each dilution. Compounds were dispensed from these solutions adding 20 μL to each well and plates were incubated for 1 h at 37 °C and 5% CO_2_. DMSO was used as a negative control and pan-PI3K inhibitor LY294002 (eight 1:2 dilutions from 50 μM to 0.391 μM, final concentrations) as a positive control for nuclear FOXO accumulation. Cells were fixed and stored as mentioned before. Cells were analyzed using the high content imaging platform Opera Phenix (PerkinElmer).

### 4.5. Data Analysis

Quantification of FOXO localization was performed by Definiens Developer v2.5 software (Definiens, München, Germany). Nucleus and cytoplasm segmentation wasdone with a custom-made ruleset using Hoechst 33342 to stain the nucleus and then expanding the area to identify the cytoplasm. After the segmentation, the ratio of green intensity was measured in both nucleus and cytoplasm and then the ratio of Nucleus versus Cytoplasm was calculated to define a threshold for translocation.

Statistics were carried out with SPSS software for calculating the percentage of FOXO translocation [35] (Jimenez et al., 2021). According to this data, the translocation ratio wasplotted in a graph against the concentration used for each compound in a logarithmic scale to approximate the EC50.

### 4.6. Immunofluorescence

In order to investigate the effect of the inhibitors on FOXO isoforms FOXO1, FOXO3 and FOXO4, 30000 U2OS or UACC62 cells were seeded on a coverslip in a 24-well plate (Falcon, Durham, NC, USA) with 500 μL medium and incubated for 24 h. Cells were treated with the compounds at 1 μM for 1 h and fixed by adding 500 μL formaldehyde 8% in PBS 1X to each well. After 10 min of incubation at RT, all volume was removed, and cells were washed with 500 μL PBS 1X and incubated 10 min at RT with permeabilization solution (1% bovine serum albumin (BSA) (Sigma-Aldrich), 0.1% Triton X-100 (Sigma-Aldrich) in PBS 1X) and then this solution was removed. Cells were washed again and incubated for 1 h at RT with blocking solution (0.1% BSA in PBS). Primary antibodies anti-FOXO1/3/4 (Cell Signaling, #2880, #2497, #9472 respectively) were prepared at 1:200 dilution in blocking solution in order to study the effect of the compounds on other FOXO isoforms. Primary antibody, 20 μL, was placed on Parafilm and the coverslip was placed upside down on the drop and incubated o/n at 4 °C. The coverslips were washed three times immersing them in PBS 1X. Secondary antibody Anti-Rabbit AF488 (Invitrogen, Waltham, MA, USA) was prepared in blocking solution at 1:200 dilution with DAPI (Invitrogen) at 1:500 dilution. The antibody drops and the coverslips were placed as before and incubated for 2 h at RT in the dark. The coverslips were washed and placed on a 10 μL Prolong (Invitrogen) drop on microscopy slides. Imaging was carried out with an optical microscope, Nikon Eclipse 90i, and photographs were taken with a DS-QiMc camera (Nikon) using NIS-Elements 3.01 software.

### 4.7. Gene Reporter Assay

To study the effect of the different compounds on the expression of FOXO target genes, 300,000 U2OS cells were seeded in a MW6 (Falcon) plate with 2 mL medium and incubated for 24 h. Transient transfection was carried out following Invitrogen’s protocol for Turbofect with pGL_6xDBE and pRL-TK plasmids. pGL3_6xDBE contains six copies of the FOXO’s DNA binding domain, DBE consensus cassette, in front of an SV40 minimal viral promoter linked to a firefly luciferase reporter gene, and pRL-TK vector expresses the *Renilla* luciferase cloned from the anthozoan coelenterate *Renilla reniformis*, controlled by the HSV-thymidine kinase promoter. After 24 h, cells were detached, centrifuged and resuspended so 20,000 cells could be seeded in a MW96 plate with 160 μL medium and incubated for 24 h. Cells were treated with 1 μM of the compounds shown in Figure 5 and incubated for 24 h. One percent DMSO was used as normalization control and 25 μM LY294002 was used as positive control. Automated luciferase assay was carried out using the Dual-Luciferase Reporter Assay System (Promega), according to the manufacturer’s instructions. Data from the readings was analyzed dividing firefly measure by *3* measure to normalize the assay by cell number, and we compared the treatment of the different compounds to negative control DMSO using Microsoft Excel software.

### 4.8. Western Blot Analysis

For the preparation of whole cell lysate, 500,000 cells seeded in 60 mm^2^-plate were harvested and lysed in lysis buffer (20 mM Tris pH 7.5, 150 mM NaCl, 1% Triton X-100, 50 mM NaF, 1 mM EDTA, 1 mM EGTA, 2.5 mM sodium pyrophosphate, 1 mM b-glycerophosphate, 10 nM Calyculin A and EDTA-free complete protease inhibitor cocktail (PIC)) (Sigma). Sample buffer was added to 1X final, and samples were boiled at 95 °C for 5 min. Samples were run on 8%–12% SDS-PAGE gels, transferred to nitrocellulose membranes and immunoblotted according to the antibody manufacturer’s instructions. Secondary antibodies were added (GE Healthcare, Chicago, IL, USA) at typically 1:10,000 dilution for 1 h at room temperature. Visualization of the signal was achieved using a ChemiDocXRS þ Imaging System (BioRad, Hercules, CA, USA).

## Figures and Tables

**Figure 1 molecules-27-05414-f001:**
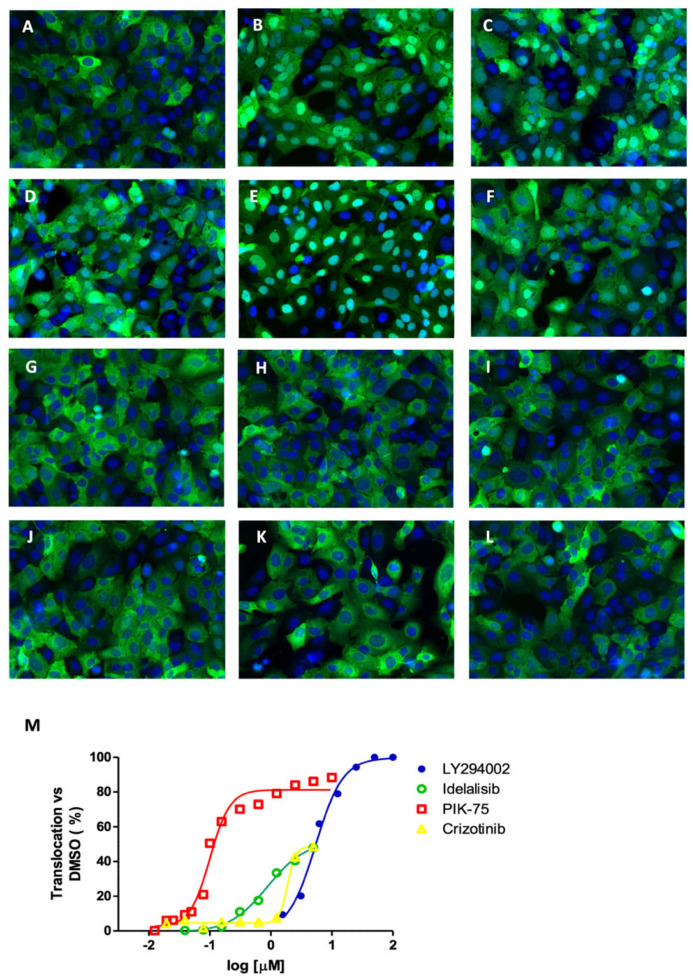
Second layer kinases effect on FOXO3 nuclear translocation. U2foxRELOC cells were treated either with (**A**) 0.5% DMSO, (**B**) 20 nM LMB, (**C**) 25 μM LY294002, (**D**) 500 nM idelalisib, (**E**) 500 nM PIK-75, (**F**) 500 nM crizotinib, (**G**) 500 nM lenvatinib, (**H**) 500 nM lapatinib, (**I**) 500 nM erlotinib, (**J**) 500 nM afatinib, (**K**) 500 nM trametinib, (**L**) 500 nM SCH772984 for 60 min. Representative images at 20× magnification are shown. (**M**) Dose-response representation of positive compounds tested in serial dilutions.

**Figure 2 molecules-27-05414-f002:**
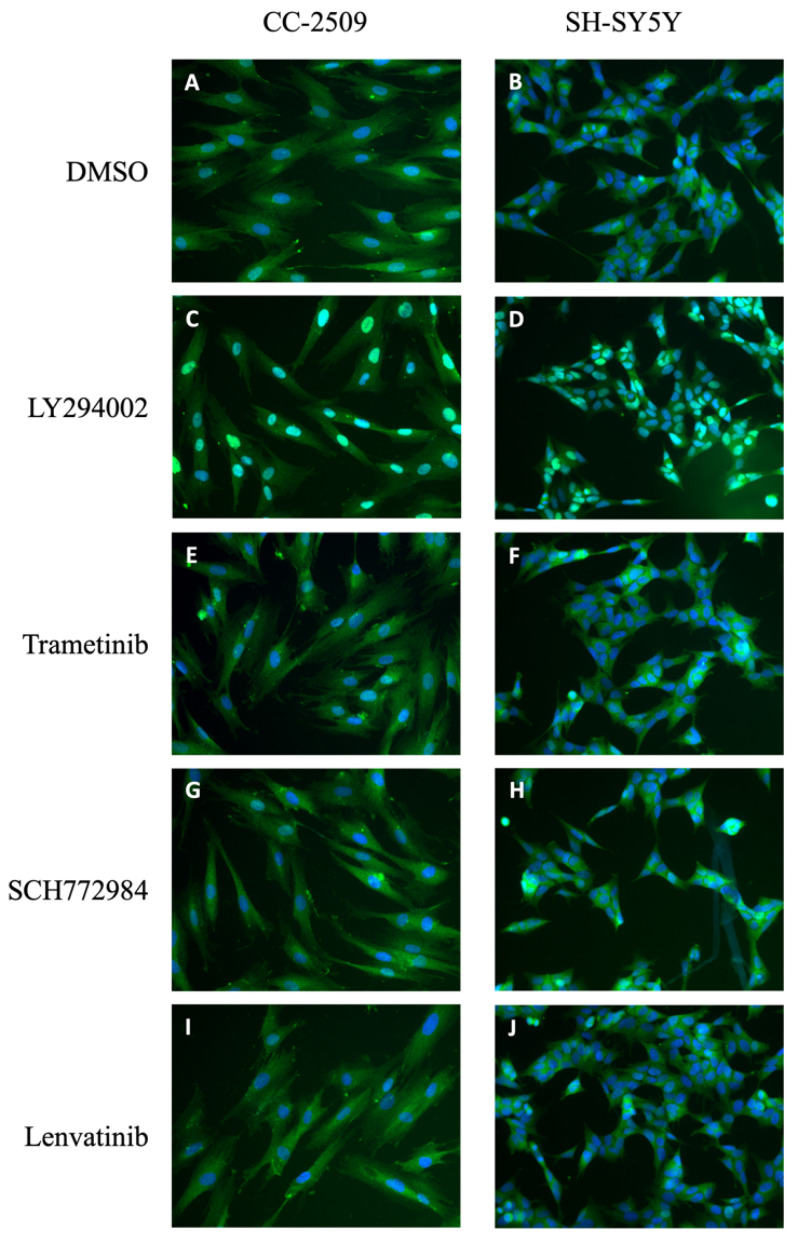
FOXO translocation in neuroblastoma cells and fibroblast. Effect on nuclear translocation of VEGFR, MEK and ERK inhibitors in CC-2509 and SH-SY5Y cells. Immunocytochemistry for FOXO3a was carried out in fibroblasts (CC-2509) and a neuroblastoma (SH-SY5Y) cell line treated for 1 h with (**A**,**B**) 0.5% DMSO, (**C**,**D**) 25 μM LY294002, (**E**,**F**) 1 μM trametinib, (**G**,**H**) 1 μM SCH772984 or (**I**,**J**) 1 μM lenvatinib. Fixed cells were incubated with green fluorescent specific antibody against FOXO3, and DAPI for nuclear staining, and imaged with fluorescence microscopy. Representative images taken at 20× magnification are shown. As it can be seen in the images taken, only LY294002 (PI3K inhibitor) causes FOXO3 nuclear translocation.

**Figure 3 molecules-27-05414-f003:**
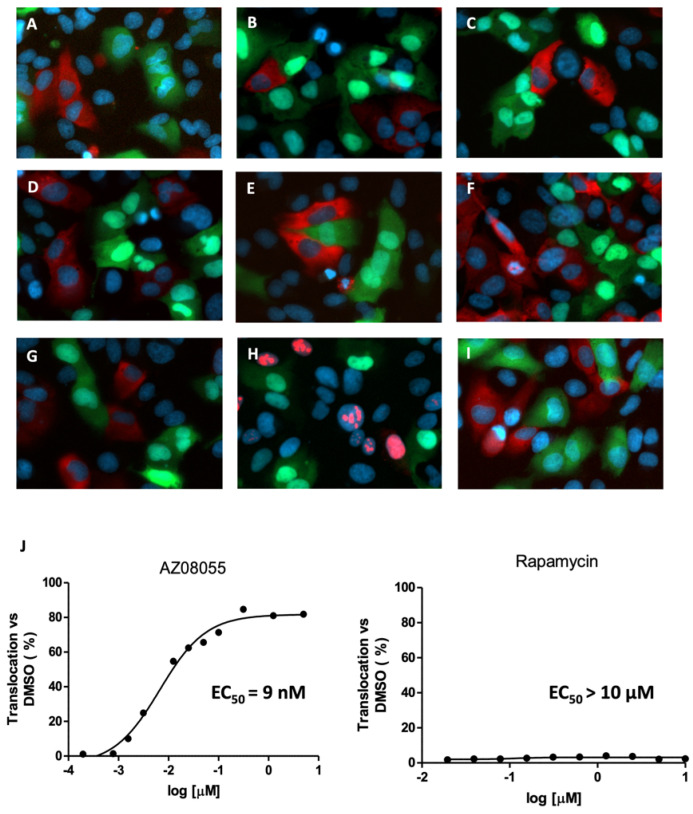
Effect of mTORC inhibitors on nuclear translocation of FOXO3. mTORC1 inhibition does not affect FOXO3 accumulation. Multiplexed assay with U2foxRELOC and U2redNES cells co-cultured and treated with (**A**) 0.5% DMSO or 500 nM, (**B**) dactolisib, (**C**) PI103, (**D**) torin, (**E**) KU-0063794, (**F**) sapanisertib, (**G**) AZD8055, (**H**) rapamycin, (**I**) 20 nM LMB for 1 h. Then cells were fixed and nuclear stained with Hoechst. Representative images are shown at 40× magnification. (**J**) Dose-response plotting of AZD8055 and rapamycin with EC_50_ of 9 nM and >10 μM, respectively.

**Figure 4 molecules-27-05414-f004:**
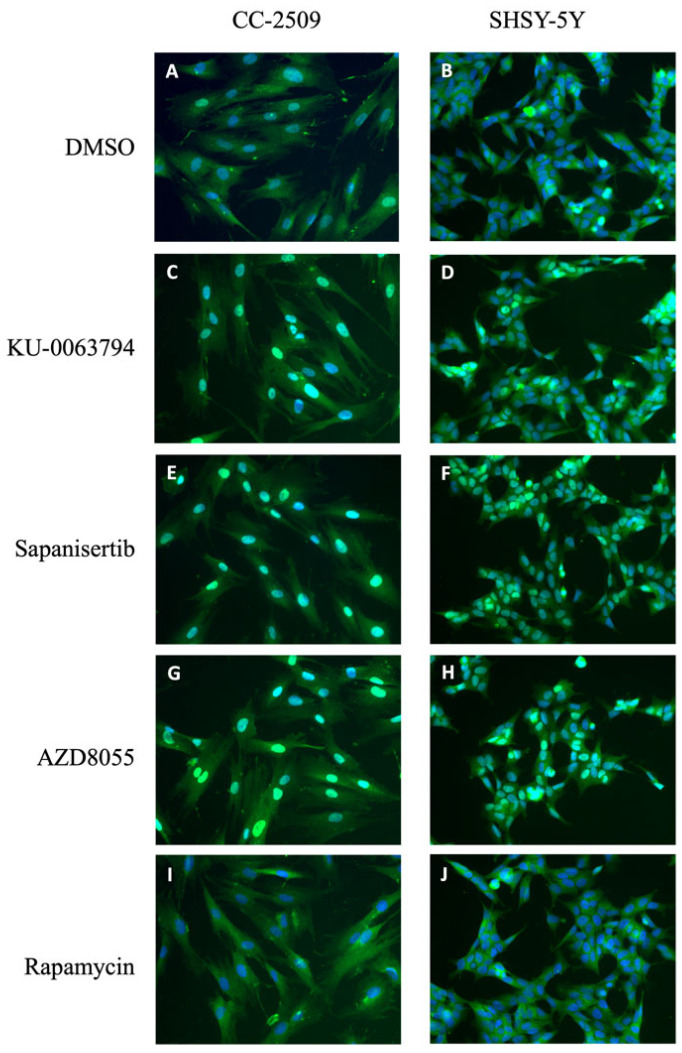
Effect of mTORC inhibitors on nuclear translocation of FOXO3 in fibroblast and neuroblastoma cell lines. mTORC1 inhibition does not affect FOXO3 localization. Immunocytochemistry for FOXO3a was carried out in fibroblasts (CC-2509) and a neuroblastoma (SH-SY5Y) cell line treated for 1 h with (**A**,**B**) 0.5% DMSO or 1 μM (**C**,**D**) KU-0063794, (**E**,**F**) sapanisertib, (**G**,**H**) AZD8055 or (**I**,**J**) rapamycin. After compound treatment, cells were fixed and incubated with a green fluorescent specific antibody against FOXO3 and DAPI for nuclear staining and imaged with fluorescence microscopy. Representative images taken at 20× magnification are shown.

**Figure 5 molecules-27-05414-f005:**
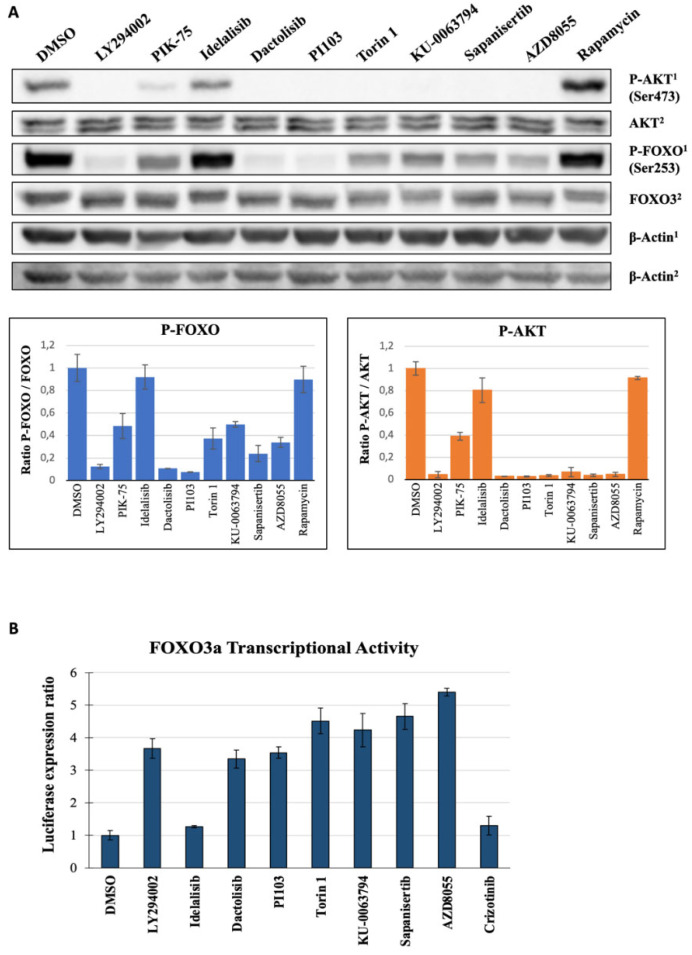
Inhibition of PI3K and mTOR affect FOXO and AKT phosphorylation and drive FOXO-dependent transcriptional activity. (**A**) U2foxRELOC cells were cultured and treated for 1 h with 1% DMSO (negative control), LY294002 25 μM (positive control) or small molecule compounds (PIK-75, idelalisib, dactolisib, PI103, torin 1, KU-0063794, sapanisertib, AZD8055, crizotinib and rapamycin) at a final concentration of 1 μM, and then were immunoblotted with specific antibodies against P-FOXO3 (Ser253) and total FOXO3, P-AKT (Ser473) and total AKT, and β-actin as the loading control. P-AKT and P-FOXO are labelled with ^1^ and should be compared with the loading control form the same membrane labelled as β-Actin^1^, while β-Actin^2^ represents the control for total AKT and FOXO3. Blots represent the quantification of P-FOXO3 and P-AKT normalized by the correspondent total protein and loading control and compared to the negative control. (**B**) U2OS cells were cultured and co-transfected with pGL_6xDBE and pRL-TK plasmids. Cells were treated for 24 h with the mentioned compounds at a concentration of 1 μM. Luciferase expression was quantified following the Dual-Luciferase Reporter Assay System. Firefly luciferase signal was normalized by *Renilla* luciferase and compared to DMSO control. Mean of triplicates and standard deviation are represented.

**Figure 6 molecules-27-05414-f006:**
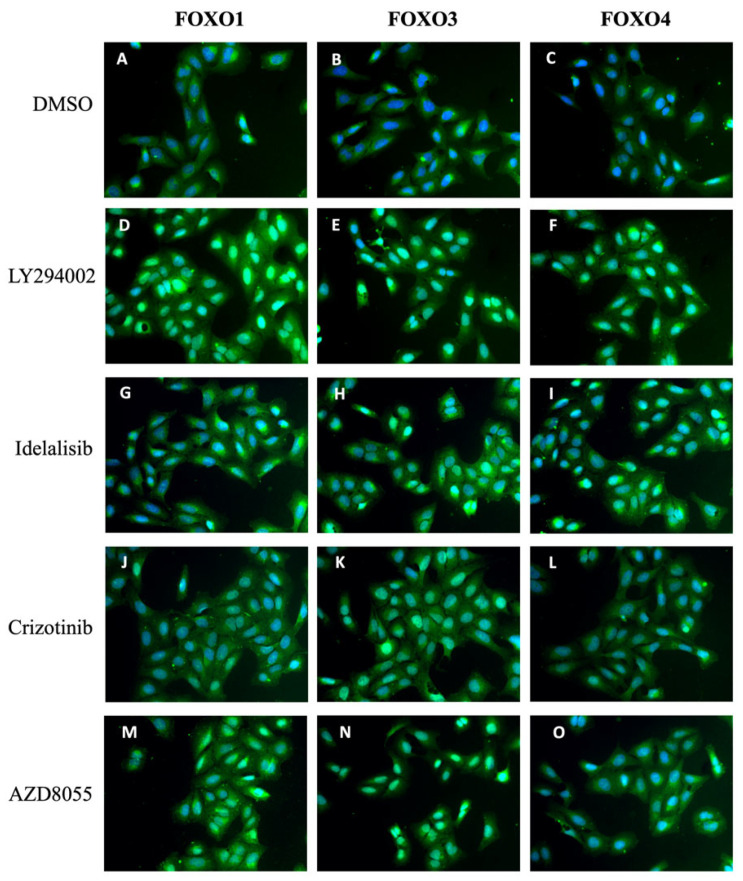
Effect of PI3K and mTOR inhibitors on FOXO1 and FOXO4 localization. U2OS cells were cultured and treated for 1 h with (**A**–**C**) 0,5% DMSO as negative control, (**D**–**F**) 25 μM LY294002 as positive control or 1 μM (**G**–**I**) idelalisib, (**J**–**L**) crizotinib, (**M**–**O**) AZD8055. Fixed cells were incubated with green fluorescent specific antibodies against FOXO1, FOXO3 or FOXO4 isoforms, and imaged with fluorescence microscopy. Representative images taken at 20× magnification are shown.

**Table 1 molecules-27-05414-t001:** Chemical compounds used to treat reporter cells.

Compound	Inhibition Target	PubChem ID
Afatinib	EGFR, HER2	10184653
AZD8055	mTORC1/2	25262965
Crizotinib	ALK, MET, ROS1	11626560
Dactolisib (BEZ235)	PI3Kα/β/γ/δ, mTOR	11977753
Erlotinib	EGFR	176870
Idelalisib	PI3Kδ	11625818
KU-0063794	mTORC1/2	16736978
Lapatinib	EGFR, HER2	208908
Lenvatinib	VEGFR1/2/3	9823820
Leptomycin B	CRM1	6917907
LY294002	PI3K (pan-inhibitor)	3973
PI103	PI3Kα/β/γ/δ, mTOR	9884685
PIK-75	PI3Kα	10275789
Rapamycin	mTORC1	5284616
Sapanisertib	mTORC1/2	45375953
SCH772984	ERK1/2	24866313
Torin 1	mTORC1/2	49836027
Trametinib	MEK1/2	11707110

## Data Availability

Not applicable.

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
