# Peer review of "mTORC2 Is the Major Second Layer Kinase Negatively Regulating FOXO3 Activity"

_molecules, 2022, doi:10.3390/molecules27175414_

Round 1

Reviewer 1 Report

The study by Jimenez et al. is a solid investigation of several lmm inhibitors of "second-layer" kinases regarding their effect on FOXO3.

I like th fact that real concentration-dependence analyses were perfomred, not just selected concentrations of inhibitors employed as frquently seen in other publications.

The following coincerns should be addressed by the authors:

The conclusions regarding effects of inhibitors of MEK etc etc missing or not missing should be discussed with respect to basal activity of the respective cascades resulting in ERK activation in the respective cell lilnes used. 

Secondly, stimulation of signaling pathways should be performed in order to assess the usefulness of inhibitors. At present, only effects of inhibitors under basal (i.e. non-stimulated) conditions are being tested.

Typo in line 131: Lack of EFFECT OF MEK...

Author Response

Thank you very much for considering the publication of our manuscript molecules-1850492 entitled “mTORC2 is the major second layer kinase negatively regulating FOXO3 activity” by Jimenez et al. in the Molecules Special Issue on "Feature Papers in Chemical Biology- Edition of 2022-2023.

            We thank both reviewers for their time and highly constructive comments who each raised several important questions that have no doubt strengthened our manuscript further. These comments in their entirety are included below and are highlighted in grey. We believe that having addressed these comments throughout our manuscript, that this revised manuscript will now be acceptable for publication.

Reviewers' comments:

Reviewer #1:

The study by Jimenez et al. is a solid investigation of several lmm inhibitors of "second-layer" kinases regarding their effect on FOXO3. I like th fact that real concentration-dependence analyses were perfomred, not just selected concentrations of inhibitors employed as frquently seen in other publications.

We thank the reviewer for this encouraging comment

The following coincerns should be addressed by the authors:

The conclusions regarding effects of inhibitors of MEK etc etc missing or not missing should be discussed with respect to basal activity of the respective cascades resulting in ERK activation in the respective cell lilnes used. Secondly, stimulation of signaling pathways should be performed in order to assess the usefulness of inhibitors. At present, only effects of inhibitors under basal (i.e. non-stimulated) conditions are being tested.

This is an excellent point raised by reviewer 1. We agree that this is an important parameter to keep in mind when analysing the inhibition of a signalling cascade. Indeed, we used cell lines that lack major activating mutations of the PI3K/AKT or MAPK pathway. U2OS, SH-SY5Y and the human dermal fibroblasts CC-2509 do not carry mutations within PI3K, AKT, PTEN, mTOR, KRAS or BRAF. However, we performed the experiments to analyse nuclear localization and the gene reporter assays in the presence of 10% FBS which contain growth factors that stimulate both PI3K/AKT and MAPK signalling. In order to unambiguously show the effect (or the lack of effect) of the MEK inhibitor trametinib in a cellular context in which the MAPK pathway is constitutively activated, we performed immunocytochemistry with UACC-62 melanoma cells that carry an activating mutation in BRAF. In addition, we performed additional experiments with U2OS cells stably expressing GFP_FOXO treated with IGF that induces signalling pathways downstream of RTK, namely the PI3K/AKT and the MAPK pathway. In both, EGF-stimulated U2OS cells and BRAF mutant UACC62 cells trametinib failed to induce nuclear FOXO3 localization indicating that MEK inhibition in the context of an acutely or constitutively activated MAPK pathway did not affect the subcellular localization of FOXO3. The new data are shown in Supplementary Figures 2 and 3, described in the result section and a section on this topic also has been added to the discussion part of the revised version of the manuscript.

Typo in line 131: Lack of EFFECT OF MEK...

We thank the reviewer for indicating this to us. We have corrected this error in the revised version of the manuscript.

Reviewer 2 Report

“mTORC2 is the major second layer kinase negatively regulating FOXO3 activity” by Jimenez et al. represents a comprehensive study that undoubtedly provides evidence of the role of second layer kinases in regulating FOXO3 activity. The author used pharmacological inhibition with clinically approved anticancer drugs to investigate their impact on the FOXO3 nuclear translocation and activity. They provided unpublished material and supplementary material to support their findings. The manuscript is acceptable for publishing in Molecules after minor changes:

1. It would be beneficial for the readers of this manuscript to discuss the contribution of observed FOXO3 nuclear translocation and activation in the overall anticancer effect of drugs that were found to be positive regulators.

2. Supplementary figures: protein bands should be labeled.

Author Response

Thank you very much for considering the publication of our manuscript molecules-1850492 entitled “mTORC2 is the major second layer kinase negatively regulating FOXO3 activity” by Jimenez et al. in the Molecules Special Issue on "Feature Papers in Chemical Biology- Edition of 2022-2023.

            We thank both reviewers for their time and highly constructive comments who each raised several important questions that have no doubt strengthened our manuscript further. These comments in their entirety are included below and are highlighted in grey. We believe that having addressed these comments throughout our manuscript, that this revised manuscript will now be acceptable for publication.

Reviewer #2:

“mTORC2 is the major second layer kinase negatively regulating FOXO3 activity” by Jimenez et al. represents a comprehensive study that undoubtedly provides evidence of the role of second layer kinases in regulating FOXO3 activity. The author used pharmacological inhibition with clinically approved anticancer drugs to investigate their impact on the FOXO3 nuclear translocation and activity. They provided unpublished material and supplementary material to support their findings. The manuscript is acceptable for publishing in Molecules after minor changes:

  1. It would be beneficial for the readers of this manuscript to discuss the contribution of observed FOXO3 nuclear translocation and activation in the overall anticancer effect of drugs that were found to be positive regulators.

We thank reviewer 2 for raising this comment. In response we included a paragraph in the revised version of the manuscript where we discuss the implication of our findings for the therapeutic effect of the approved anti-cancer drugs.

  1. Supplementary figures: protein bands should be labeled.

We thank the reviewer for indicating this to us. In order to address this comment we included a new supplementary Figure with a different order of the slots and properly labelled.